# V2G Strategies to Flatten the Daily Load Curve in Seoul, South Korea

**Sangbong Choi [1],*, Changsoo Kim [1] and Backsub Sung [2]**

[1] Power Grid Research Division, Korea Electrotechnology Research Institute (KERI),
Changwon 51543, Republic of Korea; cskim@keri.re.kr
[2] Department of Mechanical Engineering, Chosun University, Gwangju 61452, Republic of Korea;
sbsung@hanmail.net
* Correspondence: sbchoi@keri.re.kr

**Abstract:** In order to meet the increasing demand for electricity to maintain electric vehicles (EVs) worldwide, this paper aims to improve our understanding of the impact of the load on the power grid generated by the charging and discharging of electric vehicles. The rapid development of the electric vehicle (EV) industry offers new economic and environmental benefits, such as mitigating global warming by reducing carbon dioxide. On the other hand, however, we will face the reality that the emergence of such large-scale EVs will undoubtedly put additional strain on the power grid. In this context, solving the problem of excessive power usage associated with charging large electric vehicles and reducing the impact on the grid is paramount. Accordingly, in order to meet the increasing demand for electricity to maintain electric vehicles (EVs) worldwide, this paper aims to improve our understanding of the impact of the load on the power grid generated by the charging and discharging of electric vehicles. A V2G strategy is presented with the goal of flattening the daily load curve by considering the charge and the discharge positions of EVs. First, in this paper, based on the estimated share of electric vehicles, we set the assumption that EVs travel to work in the morning and leave work in the afternoon. Second, we develop an efficient V2G strategy to equalize the daily load curve due to charging and discharging of electric vehicles in Seoul by applying a system marginal price (SMP) and time-of-use (TOU) rate system. The EV charging/discharging load and existing load using V2G modeling are added up, all daily load curves are calculated and analyzed based on the 2030 and 2040 EV share scenarios for Seoul, and the grid load is leveled. The analysis suggests measures to minimize the impact of EV loads on the power grid according to the V2G strategy based on charging and discharging plans. Overall, this paper aims to smooth the grid's daily load curve and avoid grid overload by applying appropriate SMP and TOU plans; we also present an efficient V2G strategy, established through charge and discharge modeling and EV charge and discharge management techniques, in order to minimize grid expansion.

**Keywords:** V2G strategies; daily load curve; charging location; discharging location; time of use (TOU); system marginal price (SMP)

## 1. Introduction

Efforts to promote electric vehicles (EVs) and fuel cell vehicles (FCEVs) are increasing worldwide as part of the fight against air pollution caused by $CO_2$ from automobile exhaust fumes. Therefore, it is expected that the number of EVs and FCEVs connected to the distribution system will increase rapidly in the future. This expansion is expected to ultimately place a strain on utilities. Most electric power companies and governments are cooperating with each other to predict electricity demand that occurs during charging and discharging of electric vehicle batteries, support related technical standards and trade regulations, and prepare for the global spread of electric vehicles. In addition, various studies are being conducted on whether the power grid can meet the growing power demand of EVs.

To prepare for such a situation, various studies and demonstrations are being conducted on whether the power grid can meet the increasing power demand of electric vehicles using V2G technologies. V2G technology is a system technology that trades power in both directions between an electric vehicle and the power system. It not only transmits power from the power system to charge the electric vehicle but also transmits power back according to the auxiliary service market or demand response signal in the power market during power peaks. It can be expected to increase the efficiency of power production and consumption. When analyzing the trends in overseas technology in this regard, in the case of the US, a PJM-led mid-Atlantic grid-interactive car consortium and smart car consortium are proving and evaluating V2G technology, and in Korea, there is Jeju Smart Grid Demonstration Complex Smart Transportation. In the presentation business, V2G tests and demonstrations are being carried out on how to send EV (electric vehicle) batteries back to the power system. However, in order for V2G technology to achieve its final goal, it is necessary to secure large-capacity, high-output, long-life and low-cost batteries, develop communication with the vehicle's main controller and custom service to determine battery charge/discharge time, link AMI interface technology and V2G technology, and overcome V2G technology issues such as permitting and stabilizing systems for bi-directional power flow between operational vehicles and grid-connected infrastructure.

Among these various problems, grid stabilization measures using V2G strategic technology are very important, but most studies are focused on evaluating the impact of electric vehicle charging and discharging on the grid, and studies that reflect V2G strategic measures for grid stabilization are currently insufficient [1,2]. Various studies on EV energy consumption (EV battery charging load) suggest that the large-scale deployment of electric vehicles can have a significant impact on the national power grid and residential distribution system. Below, we offer an analysis of previous studies on this issue. For the distribution system in residential areas, uncoordinated charging and coordinated charging according to EV penetration were reviewed and optimization methods were presented, but charging and discharging were not considered at the same time [3]. A method for simulating electric vehicle impact on the distribution system was presented, but it was limited to charging, and no specific case review was presented [4]. Probabilistic modeling of the distribution of electric vehicles (EVs) in the power system was proposed and the resulting clustering characteristics were analyzed to predict the EV charging demand for each bus line of the distribution system, but there was no review of grid peak reduction due to electric vehicle V2G [5]. In [6], an algorithm was presented to calculate the daily load curve of electric vehicles charged by buses based on the daily charging patterns of electric vehicles according to electric vehicle supply scenarios, and to evaluate the effect of the bus load using power fluctuation analysis; however, there are downsides to this that were not taken into account. In addition, ref. [7] presented the V2G optimization technology problem for grid peak reduction, and the V2G scheduling optimization problem, that is, the problem of minimizing peaks and finding grid constraints with smart V2G technology; this seems difficult and interesting, but the problem of flattening the daily load curve was not reviewed. In [8], research focused on studying the optimal locations of electric vehicle charging stations. Also, in New York City, USA, research was conducted on how to distribute grid load and reduce costs by applying an intelligent EV charging method in order to charge the EV load during valley fill time [9]. The valley fill method is a technology that equalizes the overall load by inducing an increase in the charging load during a low-charging load-time period. Research is ongoing in Ireland to optimize EV charging cycles using demand-side management (DSM) in order to reduce both consumer costs and peak load demand on the grid [10]. In another study [11], an EV scheduling algorithm was developed to meet charging requirements of EV owners through an impact analysis of DRM (demand response management). Another study conducted in Winnipeg, Canada [12], used vehicle usage data to predict and analyze EV charging through probabilistic methods. In [13], a study was presented on minimizing peak reduction and voltage regulation problems through smart load management in residential area distribution sys-

tems. There was also a study of EV TOU (time of use) rates required to minimize the impact of EV charging on the grid. However, there are not enough studies on how to flatten the daily load curve of the metropolitan power grid using the V2G strategy of electric vehicles based on TOU and SMP rates. Several studies have explored TOU tariff system models to minimize EV charging costs in regulated markets [13] and reduce the impact of EV charging on the grid [14,15]. In ref. [16], a solution is presented for how to optimize energy management by additionally utilizing ESS to reduce electric energy bills according to the TOU pricing scheme of housing systems with PV and EV charging. Also, in ref. [17], the authors propose a charge–discharge scheduling strategy utilizing G2V/V2G and a battery energy storage system (BESS) to further improve the operation of electric vehicle charging stations. In [18], the pattern of the daily load curve according to the EV charging pattern in Seoul was analyzed, but the discharge of the EV was not considered at the same time.

Another study attempted to improve efficiency by applying an energy management strategy (EMS) to fuel cell electric vehicles (FCEVs) and hybrid electric buses (FCHEBs) for low-carbon driving [19–21]. From this perspective, most studies focus on proposing sequential charging, reducing power loss, reducing charging rates for electric vehicle owners, TOU strategies for the highest savings, and implementing EV charging stations using energy storage devices (ESSs). However, there are not enough studies based on TOU and SMP plans to flatten the daily load curve of the metro power grid by using the electric vehicle V2G strategy. Since there are few studies that have analyzed the effect of flattening the grid, it is essential to evaluate V2G strategies on how much the impact of the grid can be reduced by efficient charging and discharging of electric vehicles in the wide-area grid of a large metropolitan city. In general, EV owners want to charge and discharge their electric vehicles at work or at home, mainly because they are commuting vehicles, so it is essential to model V2G effects by place and time of charging and discharging, at home or at work.

In other words, according to V2G strategy modeling, a methodology is presented to calculate the daily charge and discharge load curve for each electric vehicle charge and discharge location in Seoul and add it to the existing load curve. More specifically, according to the Seoul electric vehicle market share scenarios in 2030 and 2040, electric vehicles are discharged when going to work in the morning, and electric vehicles are charged in the afternoon after work, which we can use as the basis for calculating the amount of electric vehicle charging and discharging. In addition, the calculation conditions are again divided into uncontrolled type (taking into account the number of vehicles operating by hour), controlled type 1, and controlled type 2 (SMP/TOU rate plan + considering the number of vehicles operating by hour), and the charging and discharging amount for each time period is calculated to calculate the daily charge and discharge for each condition. An optimization method is proposed to equalize the entire daily load curve by calculating the load curve leveling rate.

In sum, in this study, an efficient V2G strategy was developed in the direction of leveling the daily load curve of large cities according to TOU and SMP plans for large cities where electric vehicles are charged and discharged. Assuming an EV deployment scenario, this paper presents our work to apply and evaluate a method that induces electric vehicles to be discharged at work in the morning and charged at home in the afternoon. Furthermore, in this paper, a V2G strategy for flattening the load curve for a day is presented. The remainder of this paper is organized as follows. Section 2 explains grid impact analysis according to EV modeling by location and the V2G strategy as a daily load curve evaluation method. Section 3 presents the daily load curve leveling results according to the V2G strategy for each EV share scenario for Seoul in 2030 and 2040.

## 2. Daily Load Curve Evaluation Method

In this chapter, a method for evaluating daily load curves for vehicle charging and discharging at various charge and discharge locations in large cities is proposed. The charge/discharge load estimation assumes that people start charging their EVs from the grid after they get home from work in the afternoon. To mitigate grid peaks, it is also

assumed that people will discharge electricity to the grid via their vehicles after going to work in the morning.

The share of EVs was estimated based on the scenarios for the shares of electric vehicles in Seoul, Korea, in 2030 and 2040 through the long-term plan for electric vehicles in Seoul.

By combining the KEPCO (Korea Electric Power Company) SMP and TOU rates with the discharge and charge rates, the charge and discharge daily load curves for Seoul in 2030 and 2040 were calculated.

The steps for this method are described below:

(1) Calculate EV discharge probability density function at work in the morning.
(2) Calculate EV charging probability density function at home after work in the afternoon.
(3) Calculate the initial state of charge (SOC) of EV batteries at work and at home.
(4) Calculate EV charging and discharging power per hour using the charge/discharge probability density function of the EV and the SOC state of the EV battery at work and at home.
(5) Calculate the daily load curve for charging and discharging EVs in Seoul in 2030 and 2040 using the charging and discharging power of EVs based on the charge time and the discharge time either at work or at home.

Through the proposed calculation method, accurate daily load curve data were generated based on the charge and discharge load of EVs at work and at home in Seoul. In addition, a basis for establishing a billing TOU, SMP rates, and load management plans to prevent overload of the grid system in Seoul is presented.

### 2.1. Charging and Discharging Modeling of EVs by Location

For an analysis of grid impact based on the charge and discharge locations of EVs, the amount of traffic flowing in and out of downtown Seoul was analyzed based on data from the Seoul Metropolitan Police Agency's Comprehensive Traffic Information Center (Seoul Metropolitan Police Agency) [22]. To analyze the grid effect according to the charge and discharge location, the traffic volume in and out of downtown Seoul was analyzed on an annual average basis in 2020 based on data from the Traffic Information Center (Seoul Metropolitan Police Agency) of the Seoul Metropolitan Police Agency. Based on such traffic data, the difference between vehicles entering the city and those flowing out of the city centering on the Han River in Seoul was analyzed. More specifically, when analyzing the inflow/outflow of the city center, it was assumed that if there was a large amount of inflow, it would be discharged at work, and if there was a large amount of outflow, it would be charged at home. In Figure 1, a normalized value is shown.

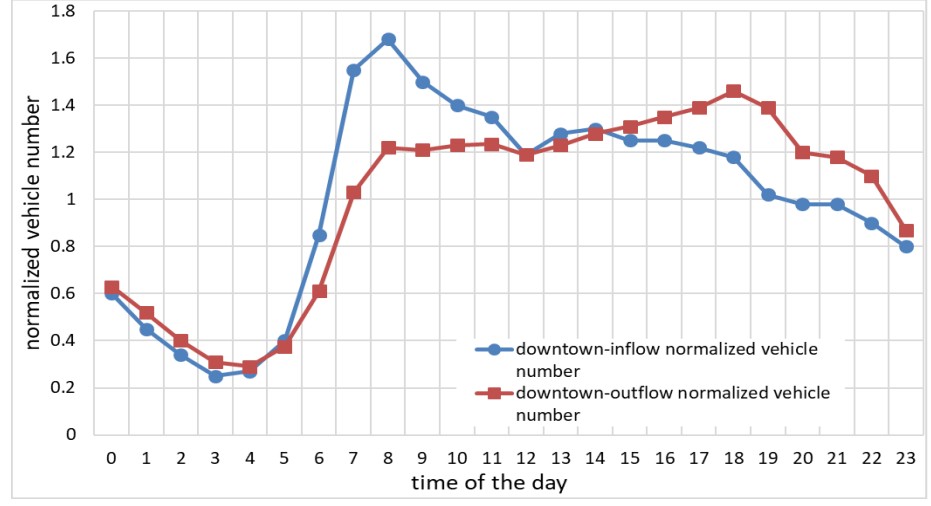

**Figure 1.** Inflow and outflow normalized traffic data for downtown Seoul.

As can be seen in Figure 1, traffic entering the city on the way to work peaks between 7:00 a.m. and 9:00 a.m., but traffic flowing out of the city after work is evenly distributed from 5:00 p.m. to midnight.

In this paper, the following assumptions are made about the inflow and outflow of vehicles in the city in order to evaluate the impact of the grid on EV charging and discharging.

(1)   Coming to work (incoming vehicle–outgoing vehicle) → EV discharge (6:00–15:00).
(2)   Leaving work (outbound vehicle–inbound vehicle) → EV charging (16:00–05:00).

Based on these assumptions, the traffic data model was used as an uncontrolled mode due to the difference between EV inflow and outflow. In addition, for the controlled mode, a model that applies both the non-controlled-mode traffic data model and the EV charging/discharging rate model was used.

Here, the SMP and TOU rate data presented in KEPCO [23] were used as the charge/discharge rate model applied to the control mode.

2.1.1. Probability Density Function of EV Discharge Start Time at the Workplace

First, it was assumed that the discharge probability density function during EV work was an uncontrolled mode calculated by subtracting the difference between the hourly outflow traffic volume and the inflow traffic volume in downtown Seoul, considering a driving time of 1 h in the morning (Equation (1)).

Second, in the case of the controlled mode, in order to minimize the effect on the system, it was calculated as in Equation (2) by considering a driving time of 1 h and applying the difference between the hourly inflow and outflow traffic in the city center and the SMP rate system. Here, the probability density function of the discharge start time at the workplace was calculated based on Equation (1), since it was assumed that the EV entering the city was used for driving to the workplace.

Therefore, it was assumed that these EVs started discharging after a certain time (e.g., 1 h) after entering the city. The probability density function at the start time of attendance in the controlled mode was calculated based on Equation (2) because the higher the SMP rate, the higher the probability that the EV would be discharged after going to work. In Equations (1) and (2), whole vehicle(t) means the entire running vehicle at time t, and $EVs_{outflow}(t)$ and $EVs_{inflow}(t)$ are, respectively, the outflow and the inflow vehicles in the city center by time zone at the actual operation time t. SMP(t) represents the SMP rate at time t.

**Equation (1)**: Probability of discharge start time at work (uncontrolled mode):

$$P_{uw}(t) = \frac{\left(EVs_{inflow}(t) - EVs_{outflow}(t)\right)}{whole\ EVs} \tag{1}$$

**Equation (2)**: Probability of discharge start time at work (controlled mode):

$$P_{cw}(t) = \frac{\left(EVs_{inflow}(t) - EVs_{outflow}(t)\right)}{whole\ EVs} \times SMP(t) \tag{2}$$

Figure 2 shows the estimated emission probability density in the uncontrolled mode at the workplace, considering the difference in traffic volume going to and from the city center. In addition, Figure 2 shows the discharge probability density results, including SMP rates for comparison with probability density using simple traffic modeling. Unlike Figures 2 and 3 presents the discharge probability density in the controlled mode in which the SMP rate is greatly adjusted during peak times. That is, as the SMP rate increases in the afternoon grid peak time period from 9:00 a.m., it can be seen that the discharge probability density increases relatively during the same time period. As shown in Figure 2, when the $SMP_2$ rate is applied rather than the $SMP_1$ rate during the same time period, the discharge probability density is high during the grid peak time period, and it can be seen that the grid peak is reduced the most.

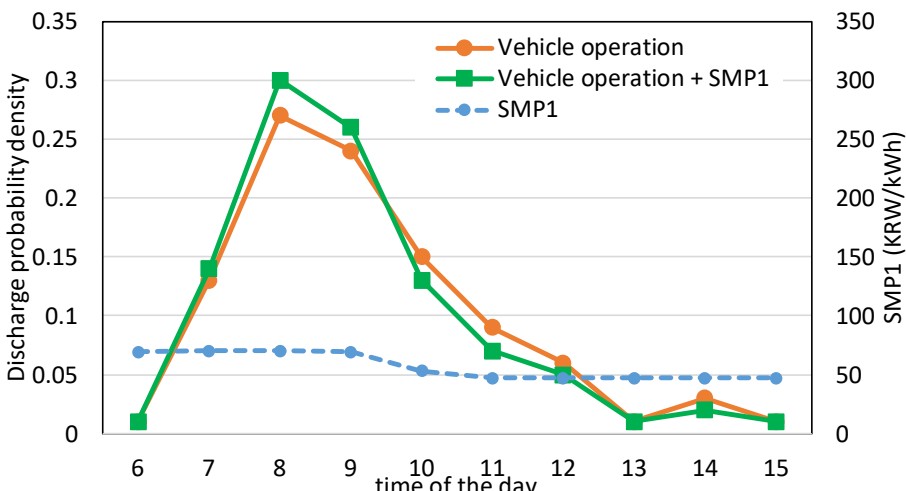

**Figure 2.** Discharge probability density in the uncontrolled mode at workplace.

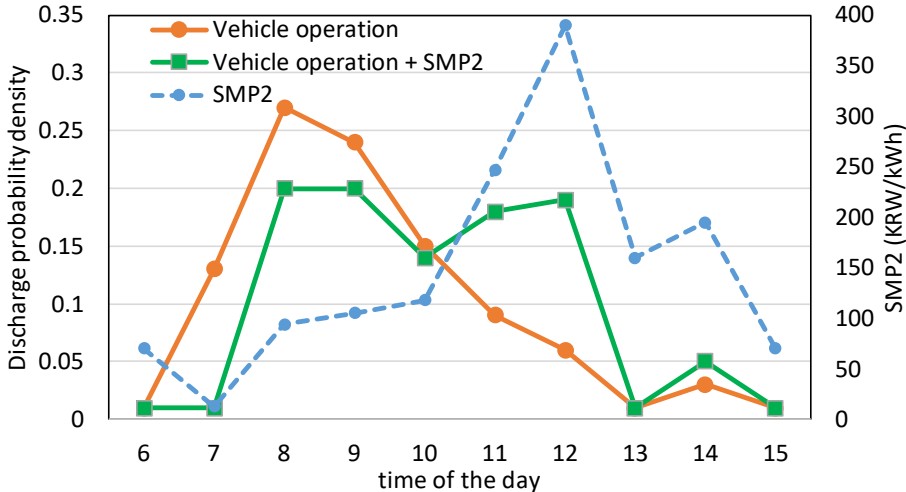

**Figure 3.** Discharge probability density in the controlled mode at workplace.

2.1.2. Probability Density Function of EV Charge Start Time at Home

As assumed before, the hourly charging probability density function for electric vehicles at home was applied to the difference between hourly inflow and outflow traffic based on a driving time of 1 h to work in Seoul in the morning, calculated for the uncontrolled mode (Equation (3)). Controlled modes for minimizing grid effects were calculated as in Equation (4) by applying the difference between inflow and outflow traffic and TOU charges over time.

The probability density function of the uncontrolled mode charge start time at home was calculated according to Equation (3) for electric vehicles departing from the city center used by owners to drive from work to home outside the city. Moreover, it was considered that these electric vehicles were likely to be charged after a certain amount of time (say, 1 h) away from the city center.

The reason why the probability density function for the controlled mode charge start time at home was calculated according to Equation (4) was that the electric vehicle was likely to be charged during the time period when the TOU charge was low after arriving at home. In Equation (3) and (4), whole vehicle(t) means the entire running vehicle at time t, and $EVs_{outflow}(t)$ and $EVs_{inflow}(t)$ are, respectively, the outflow and inflow vehicles in the city center by time zone at actual operation time t. TOU(t) represents the TOU rate at time t.

**Equation (3)**: Probability of charge start time at home (uncontrolled mode):

$$P_{uh}(t) = \frac{\left(EVs_{outflow}(t) - EVs_{inflow}(t)\right)}{whole\ EVs} \tag{3}$$

**Equation (4)**: Probability of charge start time at home (controlled mode):

$$P_{ch}(t) = \frac{\left(EVs_{outflow}(t) - EVs_{inflow}(t)\right)}{whole\ EVs} \div TOU(t) \tag{4}$$

Figure 4 shows the expected charging probability density in the uncontrolled mode at the household considering the difference between incoming and outgoing traffic. In addition, Figure 4 shows the charging probability density results including TOU rates for comparison with the charging probability density using only traffic modeling. Figure 5 shows the charging probability densities in the controlled mode where TOU rates are tuned lower during the afternoon hours. That is, as the TOU rate decreases during the off-peak hours from midnight to 5:00 a.m., it can be seen that the charging probability density increases relatively during the same time period. In particular, as shown in Figure 4, when the TOU 2 rate is applied rather than the TOU 1 rate from 6:00 p.m. to 5:00 a.m., the charging probability density is high from 6:00 p.m. to midnight, and it can be seen that the grid leveling effect is the greatest.

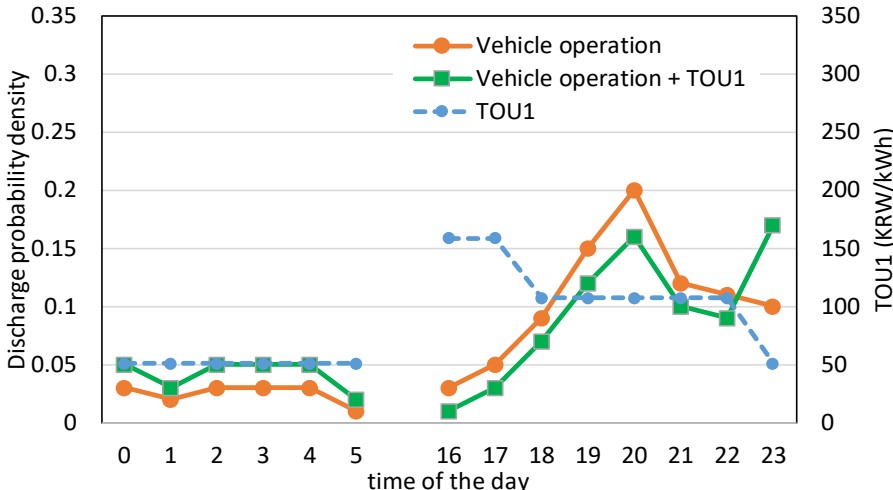

**Figure 4.** Charging probability density in the uncontrolled mode at home.

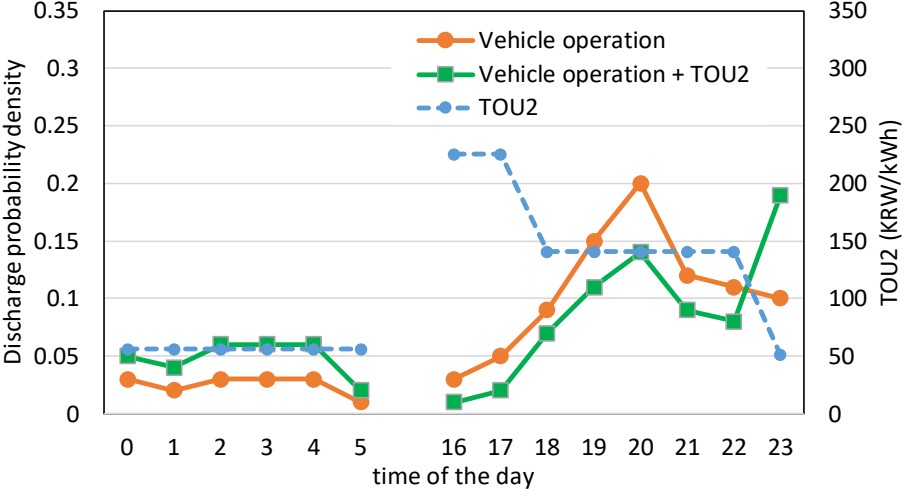

**Figure 5.** Charging probability density in the controlled mode at home.

### 2.1.3. Analysis of the Probability Density Function for Charging and Discharging EVs

As can be seen in the discharge probability density function of uncontrolled discharge and controlled discharge in the workplace, which is shown in Figure 1, in the case of the uncontrolled mode, the probability density gradually increases from 6:00 a.m. to 8:00 a.m., and then gradually decreases from 9:00 a.m. The rate of decrease, meanwhile, increases in the afternoon. In particular, the discharge probability density is at its maximum at 8:00 a.m.

In the case of control mode 1, the difference is small due to the influence of SMP deliberation, but the probability density is somewhat higher than that of the non-controlled mode. However, for control mode 2 (Figure 2), unlike control mode 1, the probability density increases from 6:00 a.m. to 7:00 a.m., increases rapidly from 8:00 a.m. to 9:00 a.m., stays almost the same until noon, and then slowly decreases. It reaches its maximum between 8:00 a.m. and 9:00 a.m. The difference between control mode 1, control mode 2, and the uncontrolled mode is caused by the fact that SMP rates are relatively high from 8:00 a.m. to 2:00 p.m.

In addition, as shown in the charging probability density function of uncontrolled discharge and controlled discharge in the home, shown in Figure 3, it can be seen that the probability density gradually increases from 4:00 p.m. to 8:00 p.m. in the case of the uncontrolled mode. After that, it gradually decreases until 11:00 p.m., and it decreases rapidly from midnight to around 5:00 a.m. In particular, the charge probability density is at its maximum at 8:00 p.m. On the other hand, the results of the controlled mode are not significantly different from those of the uncontrolled mode. However, control mode 1 shows a relatively low afternoon charging probability density and a slightly higher charging probability density from midnight to 5:00 a.m. due to the influence of the TOU tariff. The reason is that TOU tariffs are very low from 11 p.m. to dawn. However, as shown in Figure 3, in the case of control mode 2, it can be seen that the probability density is slightly lower in the afternoon and slightly higher in the morning than in control mode 1.

Figures 6–8 show the EV charge/discharge density function calculated through the uncontrolled mode using only vehicle inflow and outflow modeling, and control mode 1 and control mode 2 modeling using the vehicle's SMP rate and TOU rate system, respectively. Analyzing Figures 6–8, it can be seen that the discharge probability is high in the grid off-peak time zone, and that the charge probability is high in the peak time zone in the uncontrolled mode compared to the controlled mode. In addition, it can be seen that the case of control mode 2 has a higher charging probability in the grid off-peak time zone and a higher discharge probability in the peak time zone compared to control mode 1. Therefore, it can be seen that control mode 2 is most effective for grid leveling.

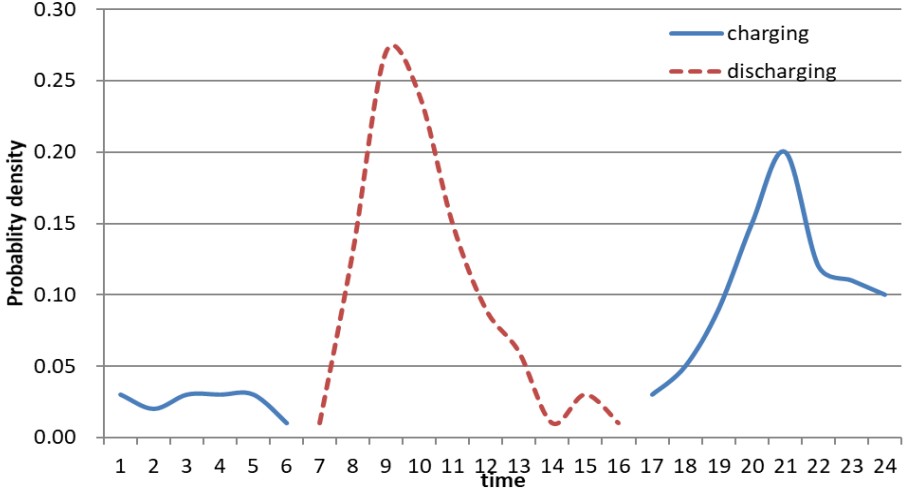

**Figure 6.** Charging and discharging probability density function by hour for the uncontrolled mode (vehicle operation).

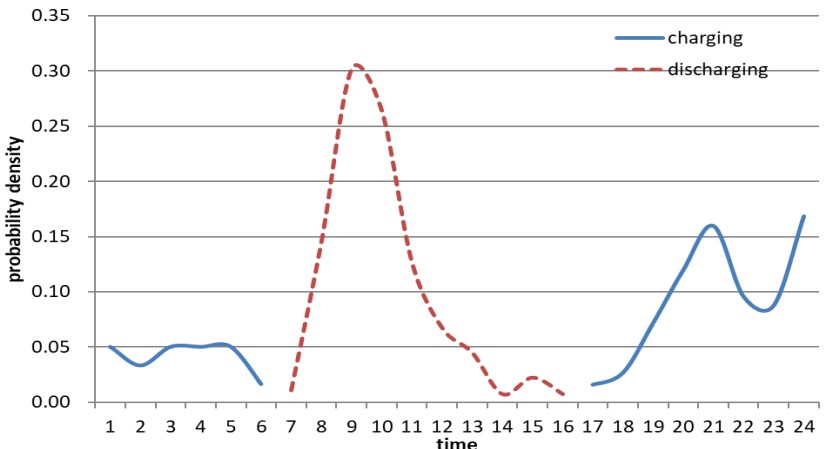

**Figure 7.** Charging and discharging probability density function by hour for control mode 1 (vehicle operation + $SMP_1$, $TOU_1$ tariff).

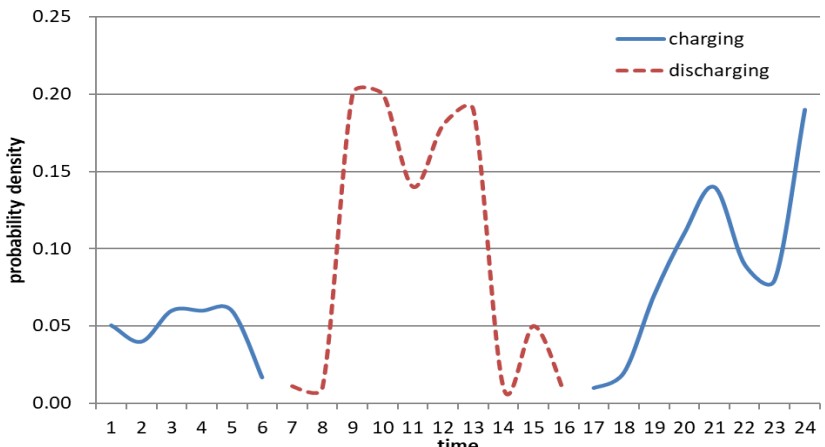

**Figure 8.** Charging and discharging density function by hour for control mode 2 (vehicle operation + $SMP_2$, $TOU_2$ tariff).

### 2.1.4. Calculating the SOC Condition of the EV Battery

The initial battery SOC at the time of going from home to work and vice versa is calculated using Equation (5), which considers the mean driving distance predetermined for EV charging:

$$soc_0 = \left(1 - \frac{\alpha}{d_R}\right) \times 100\% \tag{5}$$

where $\alpha$ is the daily driving distance and $d_R$ is the EVs' maximum driving distance.

A daily driving distance of 46.2 km from home to work and vice versa was used to calculate the initial battery $SOC_0$ based on a charging station in Seoul using Equation (5). The distance was obtained from the 2010 Vehicle Driving Distance Analysis Report published by the Seoul Metropolitan Police Agency [24]. Meanwhile, the maximum driving distance of the EVs was set to 180 km according to the EVs' efficiency of 0.16 kWh/km when using a Nissan Altra 29 kWh lithium-ion battery [25]. We used constant values for the initial $SOC_0$ and charging times for each electric vehicle because we applied the average values for evaluating the daily load curves for charging electric vehicles on a large scale, such as a big city.

Therefore, based on the 46.2 km daily driving distance from home to work and vice versa, the initial battery $SOC_0$ at the time of starting from home to work and vice versa was calculated as 74%. Therefore, the initial battery $SOC_0$ after 2 days of driving would be 0.49 if the initial battery $SOC_0$ was high enough not to require recharging after 1 day of driving

from home to work and vice versa, and it would need recharging after 2 days of driving the same distance.

Accordingly, the charging time Tc(h) in the workplace and at home for the Nissan Altra lithium-ion battery could be calculated using Equation (6). In Equation (6), the initial battery state of charge = $SOC_0$ and the final battery state of charge = $SOC_F$. In summary, the EV battery $SOC_F$ was set to 1, indicating a full charge. In contrast, the hourly battery charging power (Bc) was set to 6.0 kW considering the hourly charging power characteristics of the Nissan Altra lithium ion-battery. Therefore, Tc(h) at the workplace and at home was determined to be 5.0 h. The initial $SOC_0$ and the charging time of all EVs in Equations (5) and (6) were the same because it was assumed that all vehicles had the same initial $SOC_0$ and average mileage for analysis purposes of the daily load curve of EVs charging on a large scale, such as in a big metropolitan area.

$$T_c = \frac{(SOC_F - SOC_0)}{B_C} \times 29 \text{ kWh} \tag{6}$$

### 2.2. Grid Impact Analysis Using EV Charge and Discharge Modeling through V2G Strategy

In Section 2.1, charge/discharge model algorithms for each location at home and work and charge/discharge probability density functions for controlled and uncontrolled modes were proposed. That is, the daily charge/discharge load curve of the electric vehicle battery was calculated considering the charge/discharge probability density function, the initial SOC state of the electric vehicle, the charge/discharge time, and the charge/discharge power per hour.

The number of electric vehicles was calculated by setting the shares of EVs to 10% and 30% of total vehicles in 2030 and 2040, respectively. Specifically, through the V2G charging and the discharging strategy of the uncontrolled mode and controlled mode, the partial load for charging and discharging of one EV was first calculated, and this value was applied to the total number of electric vehicles for the year in order to calculate the EV charge and discharge load curve.

In addition, the existing daily load curve in Seoul was calculated based on the results of the load pattern analysis of the acceptance standard of the Korea Energy Management Corporation in 2018 [26]. Figure 9 shows the result of calculating the conventional daily load curve.

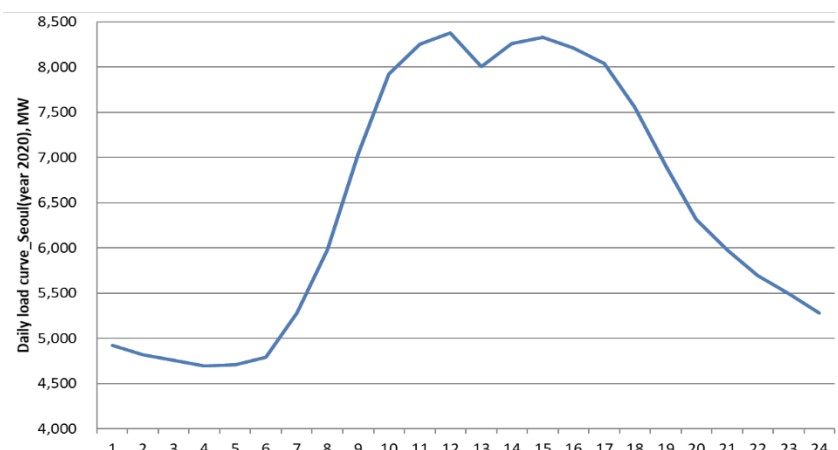

**Figure 9.** Existing daily load curve in Seoul, South Korea.

Based on Figure 9, a regression analysis method was applied to calculate the existing load curves for Seoul in 2030 and 2040. In addition, considering the existing load curve and the V2G charge/discharge strategy curve, which reflects the electricity rate system presented in Figures 6–8, the entire partial subsurface curve for Seoul was calculated. A detailed overall flowchart is shown in Figure 10 below.

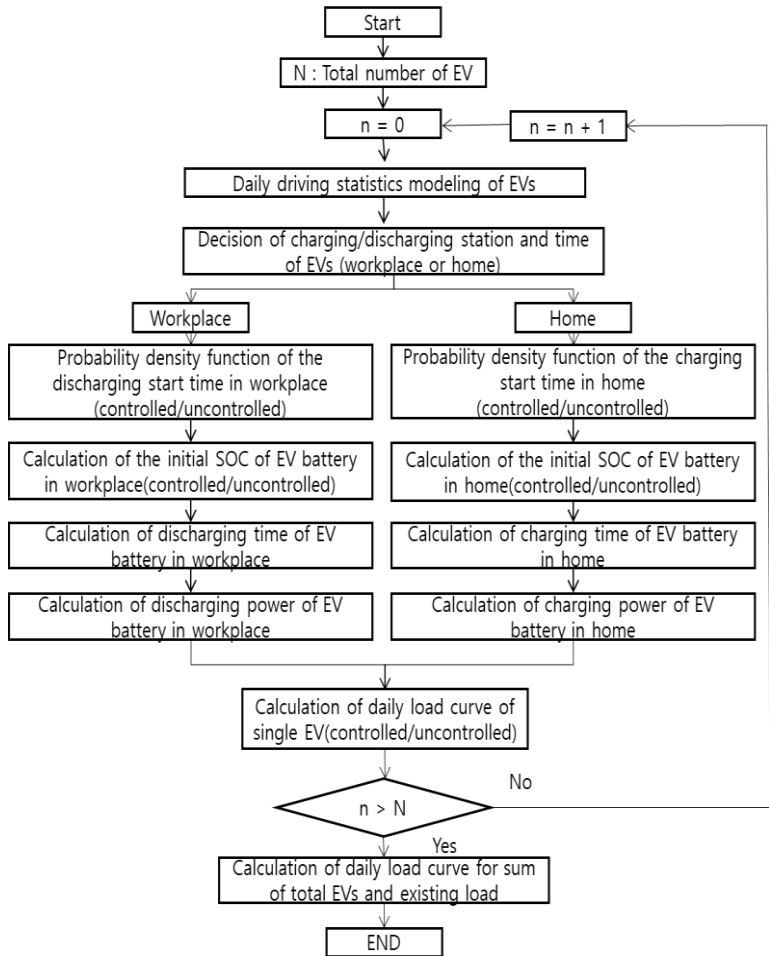

**Figure 10.** Flowchart of daily load curve calculation through V2G strategy in Seoul, Korea.

According to the flowchart shown in Figure 6, the total load curve was calculated by adding the existing loads in Seoul in 2030 and 2040 and the electric vehicle charge and discharge loads, calculated through the uncontrolled mode, control mode 1, and control mode 2.

Figures 11–16 show the total load curves for Seoul when adding the V2G daily load curve to the existing daily load curve.

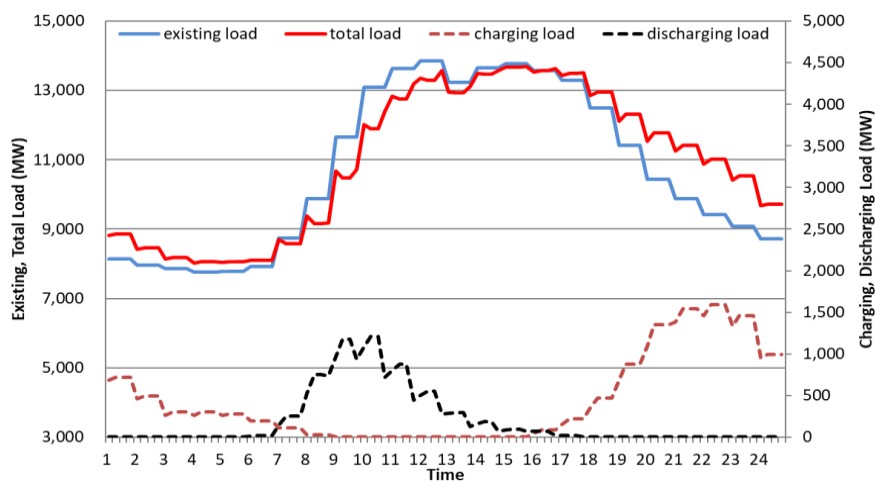

**Figure 11.** Daily load curve in Seoul in 2030 (uncontrolled mode).

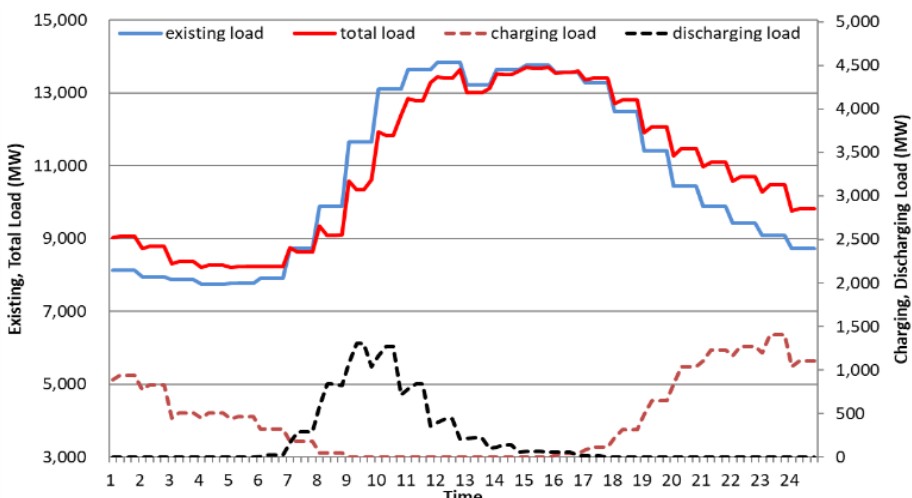

**Figure 12.** Daily load curve in Seoul in 2030 (controlled mode 1).

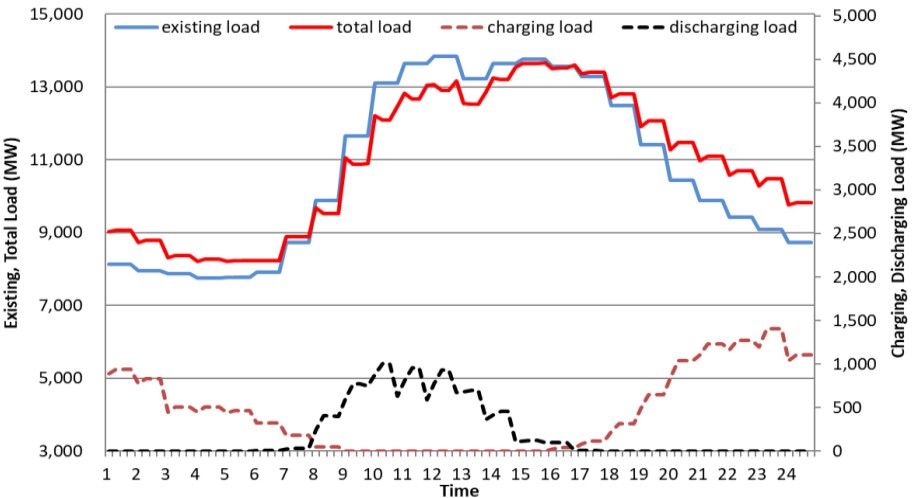

**Figure 13.** Daily load curve in Seoul in 2030 (controlled mode 2).

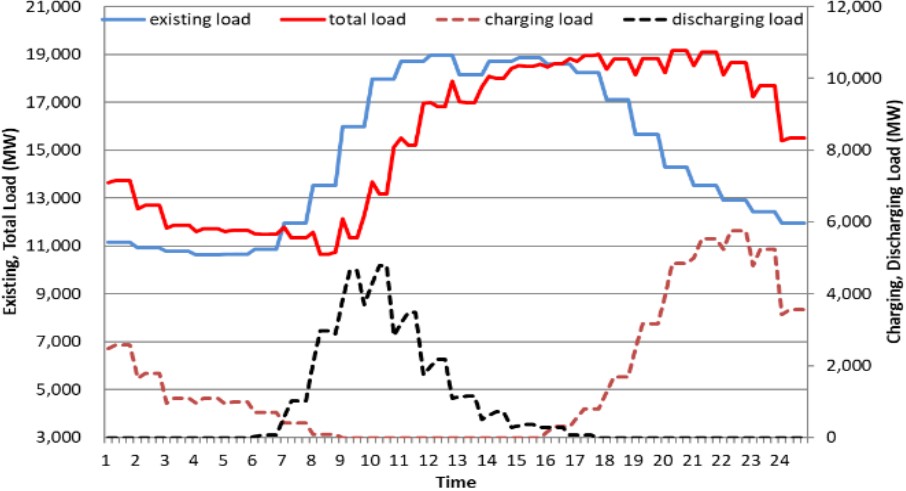

**Figure 14.** Daily load curve in Seoul in 2040 (uncontrolled mode).

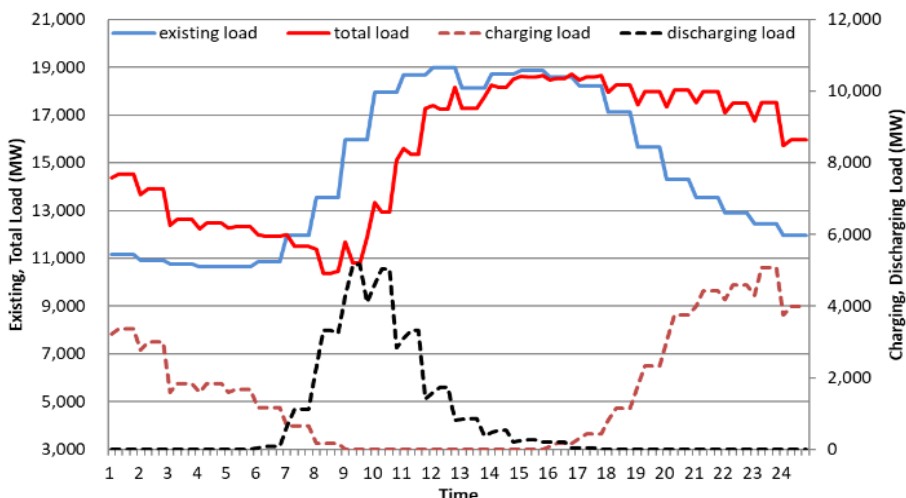

**Figure 15.** Daily load curve in Seoul in 2040 (controlled mode 1).

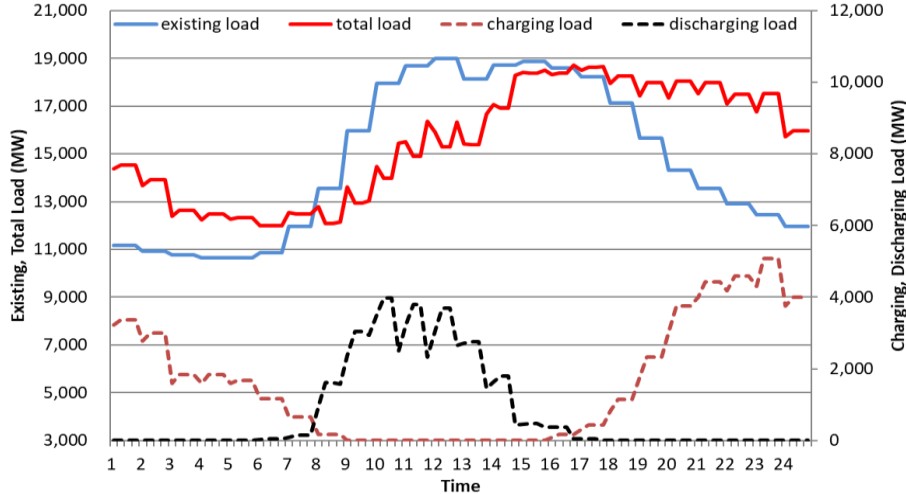

**Figure 16.** Daily load curve in Seoul in 2040 (controlled mode 2).

As shown in Figures 11–16, the conventional 2030 and 2040 daily load curves indicate that the load gradually increases from 9:00 a.m., decreases from 12:00 a.m. to 1 p.m., increases from 1 p.m., reaches its peak around 2 p.m., and then gradually decreases from 6 p.m. In the case of the uncontrolled mode, the discharging starts at 6:00 a.m. (discharging time at the workplace), and the load gradually increases from that time. It flattens at 9:00 a.m., and then it decreases. In the case of control mode 1, the load curve shows a similar behavior to the uncontrolled mode, but the discharge load is higher due to the SMP tariff. Control mode 2 is different from control mode 1 in that the discharging load stays high and continues until the grid peak. Overall, the controlled mode associated with the SMP fare has a lower discharge rate in the morning and a higher discharge rate in the afternoon than the uncontrolled mode.

## 3. Conclusions

According to the estimation of the shares of electric vehicles in Seoul in 2030 and 2040, the amounts of electric vehicle charge and discharge for the corresponding years were calculated. More specifically, considering whether or not the charging and discharging rate system was reflected, the daily load curve for charging and discharging electric vehicles for each year was calculated by dividing the scenarios into an uncontrolled mode, control mode 1, and control mode 2.

When calculating daily load curves for the uncontrolled mode, control mode 1, and control mode 2 in 2030 and 2040 for Seoul, compared to the uncontrolled mode considering only vehicle operation, the current charge and discharge rates applied by KEPCO [23] were reflected. It was found that control mode 1 flattened the daily load curve more, and control mode 2 (to which the more adjusted charge/discharge rate was applied) flattened the daily load curve the most. Putting these together, the following conclusions can be derived.

First, when the EV share is 10% (in 2030), neither the controlled mode nor the uncontrolled mode show high charge and discharge loads at home or in the workplace. Concerning the uncontrolled mode, the total load increases by 1–2% from late at night to 10:00 a.m. (off-peak time). Between 11 a.m. and 5 p.m. (peak time), it decreases by 1–4% for control mode 1 and control mode 2. Therefore, applying the TOU and SMP tariffs can increase the morning load and decrease the load in the afternoon.

Second, when the EV share is 30% (2040), control mode 1 and control mode 2 show high charge and discharge loads at home and in the workplace. In the case of control mode 1, the total load increases by 2–3% from late at night to 10 a.m. (off-peak time) and decreases by 3–9% between 11 a.m. and 5 p.m. (peak time). In the case of control mode 2, it increases by 3–5% from late at night to 10 a.m. (off-peak time) and decreases by 6–15% between 11 a.m. and 5 p.m. (peak time).

To conclude, the TOU and SMP tariffs could significantly increase the morning load and decrease the load in the afternoon. Control mode 2 can flatten the load curve most effectively compared with the other modes.

The results of this paper also provide an accurate picture of how much the V2G strategy contributes to daily load flattening via the charge and discharge loads of electric vehicles at workplaces and homes in Seoul. In other words, this paper proposes a strategy for establishing an SMP rate and a TOU rate management plan in order to flatten the daily load curve of the Seoul power system and prevent overload through a V2G strategy. However, research to establish a V2G strategy that simultaneously considers EVs and FCEVs will be conducted in the future, factoring in the influence of other external factors such as seasonal demand changes and energy supply and demand.

**Author Contributions:** Conceptualization, S.C.; Validation, B.S.; Formal analysis, C.K. All authors have read and agreed to the published version of the manuscript.

**Funding:** This research was funded by the Korea Institute of Energy Technology Evaluation and Planning (Grant No. 20192010107050).

**Institutional Review Board Statement:** Not applicable.

**Informed Consent Statement:** Not applicable.

**Data Availability Statement:** Data are not publicly available due to privacy or ethical restrictions.

**Conflicts of Interest:** The authors declare no conflict of interest.

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
