# Peer review of "V2G Strategies to Flatten the Daily Load Curve in Seoul, South Korea"

_applsci, doi:10.3390/app131810392_

Round 1
Reviewer 1 Report (Previous Reviewer 2)
The revised manuscript has been greatly improved, but there are still some minor problems. It is suggested that the author make modifications to improve the quality of the manuscript:
(1)Recommending the author to conduct a thorough review of the manuscript format and implement necessary modifications.
(2)The quality of the images in Figure 6-9 and Figure 11-16 is subpar. It is recommended that the author make modifications to them as demonstrated in Figure 2-5.
The English can be slightly adjusted following inspection.
Author Response
(1) Recommending the author to conduct a thorough review of the manuscript format and implement necessary modifications.
- As you pointed out, we thoroughly reviewed the format of the manuscript and made any necessary corrections and specific corrections are marked in red in the paper.
(2) The quality of the images in Figure 6-9 and Figure 11-16 is subpar. It is recommended that the author make modifications to them as demonstrated in Figure 2-5.
- As pointed out, the image quality in Figure 6-9 and Figure 11-16 was increased to match the image quality level shown in Figure 2-5. Specific corrections are marked in red in the paper.
- English editing was performed through the MDPI English service.
- Thank you for your kind review

Reviewer 2 Report (Previous Reviewer 3)
The authors corrected all my comments.
Author Response
- we thoroughly reviewed the format of the manuscript and made any necessary corrections. Specific corrections are marked in red in the paper.
- English editing was performed through the MDPI English service.
- Thank you for your kind review

Reviewer 3 Report (Previous Reviewer 4)
This article explores the additional load caused by electric vehicles on the power grid and proposes a solution to smooth the daily load curve of the power grid using the V2G strategy.
The article has been revised to meet the publication requirements of the journal, but there are still several issues that need to be addressed.
1. There is too much content in the abstract section. It is recommended that the author condense the abstract.
2. In the introduction section, to solve the problem of carbon dioxide pollution in car exhaust, measures taken globally not only promote electric vehicles, but also include the development of fuel cell vehicles. Therefore, it is necessary to introduce fuel cell vehicles in the introduction. It would be beneficial to incorporate the subsequent citation into the article.:
https://doi.org/10.3390/su14106320
https://doi.org/10.1016/j.energy.2023.127105
https://doi.org/10.1016/j.energy.2023.128462
3. The last paragraph of the introduction should provide a detailed description of the author's work on the content of the article.
4. The sizes of Figures 11-16 are inconsistent.
5. The conclusion section can provide a more detailed description of the next work.
It is recommended that the author carefully revise the above comments.
The English quality of the article is high, using rigorous academic language, presenting clear research objectives, content, and methods, and presenting clear research conclusions.
Author Response
(1) There is too much content in the abstract section. It is recommended that the author condense the abstract.
- As you pointed out, the abstract part was shortened and modified. Specific corrections are marked in red in the paper.
(2) In the introduction section, to solve the problem of carbon dioxide pollution in car exhaust, measures taken globally not only promote electric vehicles, but also include the development of fuel cell vehicles. Therefore, it is necessary to introduce fuel cell vehicles in the introduction. It would be beneficial to incorporate the subsequent citation into the article.
- As pointed out, in the introduction part, the issue of encouraging the spread of electric vehicles and fuel cell electric vehicles was raised to solve the problem of carbon dioxide pollution caused by automobile exhaust gases, and the need to introduce FCEV was included. We have also reviewed the article you provided and included it as a reference. Specific corrections are marked in red in the paper.
(3) The last paragraph of the introduction should provide a detailed description of the author's work on the content of the article.
- In the last paragraph of the introduction, a detailed explanation of the work related to the content of the article was provided. Specific corrections are marked in red in the paper.
(4) The sizes of Figures 11-16 are inconsistent.
- The size of Figure 11-16 has been modified to match. Specific corrections are marked in red in the paper.
(5) The conclusion section can provide a more detailed description of the next work.
- In the conclusion section, it was added that future research will be conducted on V2G strategies by season and energy supply and demand, including EV and FCEV. Specific corrections are marked in red in the paper.
- English editing was performed through the MDPI English service.
- Thank you for your kind review

Round 2
Reviewer 3 Report (Previous Reviewer 4)
After careful editing by the author, the paper has reached the level of journal publication.
The English proficiency of the article has reached the level of journal publication.
This manuscript is a resubmission of an earlier submission. The following is a list of the peer review reports and author responses from that submission.
Round 1
Reviewer 1 Report
The paper provides an overview of possible scenarios in 2030 and 2040 of daily load in Seoul considering charging and discharging of Plug-in Hybrid Electric Vehicles to the grid.
The reviewer would positively comment on the broad area of interest of this research, however major revisions are necessary to enhance the quality of this study.
A detailed transcript of needed improvements are listed below for the authors’ convenience.
· As a general comment, different typos can be found throughout the paper. Please, proofread the article in its entirety and correct these errors to improve readability.
· A similar comment can be done at the presentation level, chapter 2 title is at the end of page 2 after the footnote while it should be directly in the next page; some values inside Table 1 are not aligned; figures quality is very low as it seems that some of them were saved using screenshots. Please, revise all these comments and improve the quality of the presentation.
· A last general comment can be done on the figures. All of them are done in Excel which might not be the best tool for preparing graphs for a journal paper. Consider using a different software, if possible.
· Specifically on the content of the research, the reviewer would suggest a thorough literature review especially on some strong sentences such as the one at the beginning of page 2 where references 3 to 7 are cited. These citations are very old, one of them is dated 1973 and the reviewer believes that the state-of-the-art should be presented, not what was written 5 decades ago.
· The reviewer would suggest to include a breakdown of the next chapters at the end of the introduction, to guide the reader through the study.
· Some acronyms are not first explained, such as KEPCO that appears firstly in page 3, or a reference to a certain “Valley Fill” time in page 2 without any further explanation.
· Avoid repetitions if possible, such as at the beginning of page 3 where “charging and discharging” was repeated 3 times.
· In fig. 1, the same data is repeated twice as shown in the legend. Please, change the label of the x-axis to make it more understandable. A suggestion would be “Time of the Day”.
· The same data shown in Fig. 1 are not well explained in the text, where only a reference to a certain study is done. Please, explain where these data come from and if they are an average over a certain period of time or anything else that may help the reader.
· Most of the time equations present terms that are not defined anywhere in the text, please revise them including an explanation.
· Tables 1 to 4 are not helpful to the reader, the numbers do not make the study more understandable. Try representing these values in different ways, maybe using graphs.
· In equation 5, there is a reference to State of Charge but then an energy is reported. The reviewer would suggest to refer to it as energy in the battery since SOC is usual expressed as percentage or fraction.
· The reviewer is concerned on a major assumption that was done in this research, namely that the SOC could drop to 10% while discharging at the workplace. It is concerning because such low levels would age the battery very fast and require a new one in short time.
· It is not clear to the reviewer if this paper consider PHEVs or EVs, since in the introduction they seem the same thing while they are not. If the study considers PHEVs it is important to understand that these vehicles can be charged using the internal combustion energy and due to their small battery capacity having a charge-discharge on a daily basis would “kill” the battery. This point is really important to clarify.
· In page 12 the authors state that “control mode 2 is the most efficient for grid peak reduction” but no analysis or numbers are seen. Please, support such statement with numbers that confirm what you say.
English can be improved but main focus is on typos and presentation.
Author Response
(1) As a general comment, different typos can be found throughout the paper. Please, proofread the article in its entirety and correct these errors to improve readability.
- Improved readability by correcting typos and errors throughout the thesis.
(2) A similar comment can be done at the presentation level, chapter 2 title is at the end of page 2 after the footnote while it should be directly in the next page; some values inside Table 1 are not aligned; figures quality is very low as it seems that some of them were saved using screenshots. Please, revise all these comments and improve the quality of the presentation.
- Moved Chapter 2 title to the next page and improved the number quality.In addition, all tables related to the probability density function have been replaced with figures, and the quality of the figures has been improved.
(3) A last general comment can be done on the figures. All of them are done in Excel which might not be the best tool for preparing graphs for a journal paper. Consider using a different software, if possible.
- Improved the picture quality by using the picture to be applied in the thesis as the most suitable tool.
(4) Specifically on the content of the research, the reviewer would suggest a thorough literature review especially on some strong sentences such as the one at the beginning of page 2 where references 3 to 7 are cited. These citations are very old, one of them is dated 1973 and the reviewer believes that the state-of-the-art should be presented, not what was written 5 decades ago.
- As you pointed out, we presented a more specific literature review for references 3-7 and added new literature, excluding older citations.However, please understand that there are not many references related to grid peak reduction using V2G, so many references cannot be reviewed.
(5) The reviewer would suggest to include a breakdown of the next chapters at the end of the introduction, to guide the reader through the study.
- As pointed out, the analysis of the next chapter is presented at the end of the introduction.
(6) Some acronyms are not first explained, such as KEPCO that appears firstly in page 3, or a reference to a certain “Valley Fill” time in page 2 without any further explanation.
- The abbreviation for KEPCO, a Korean electric power company, has been explained and an explanation of the valley fill method has been added.
(7) Avoid repetitions if possible, such as at the beginning of page 3 where “charging and discharging” was repeated 3 times.
- As you pointed out, I've avoided charging and discharging loops at the beginning of page 3 as much as possible.
(8) In fig. 1, the same data is repeated twice as shown in the legend. Please, change the label of the x-axis to make it more understandable. A suggestion would be “Time of the Day”.
- As you pointed out, I modified the label of the x-axis to be Time of the Day
(9) The same data shown in Fig. 1 are not well explained in the text, where only a reference to a certain study is done. Please, explain where these data come from and if they are an average over a certain period of time or anything else that may help the reader.
- As pointed out, the source and period were presented for the data in Figure 1, and an explanation was added.
(10) Most of the time equations present terms that are not defined anywhere in the text, please revise them including an explanation.
- For the proposed equations, terms have been suggested and explanations have been added.
(11) Tables 1 to 4 are not helpful to the reader, the numbers do not make the study more understandable. Try representing these values in different ways, maybe using graphs.
- As pointed out, Tables 1 to 4 are presented as graphs instead.
(12) In equation 5, there is a reference to State of Charge but then an energy is reported. The reviewer would suggest to refer to it as energy in the battery since SOC is usual expressed as percentage or fraction.
- For Equation 5, the SOC was expressed as a percentage, the formula was modified, and a specific explanation was provided.
(13) The reviewer is concerned on a major assumption that was done in this research, namely that the SOC could drop to 10% while discharging at the workplace. It is concerning because such low levels would age the battery very fast and require a new one in short time.
- When simulating in the presented thesis, charging and discharging were considered so that the SOC does not fall below 10% in reality, so it is judged that battery aging will not be a big problem.
(14) It is not clear to the reviewer if this paper consider PHEVs or EVs, since in the introduction they seem the same thing while they are not. If the study considers PHEVs it is important to understand that these vehicles can be charged using the internal combustion energy and due to their small battery capacity having a charge-discharge on a daily basis would “kill” the battery. This point is really important to clarify.
- As you pointed out, this paper considered EV, not PHEV, and PHEV terms were not used throughout the paper.
(15) In page 12 the authors state that “control mode 2 is the most efficient for grid peak reduction” but no analysis or numbers are seen. Please, support such statement with numbers that confirm what you say.
- As you pointed out, we have additionally suggested that control mode 2 is the most efficient in reducing grid peaks (i.e., compared to uncontrolled mode, control mode 1, it increases the total load by 2% during morning off-peak as of 2040, By reducing the total load by more than 6% during peak, it was explained in the conclusion that it contributed the most to the leveling of the curve)
Detailed modifications are marked in red in the attached file.

Reviewer 2 Report
The authors should carefully check the format, quality of figures and other issues in writing.
1、 There are many grammatical errors in the manuscript. The author should check and revise carefully.
2、 The manuscript uses so many long sentences that the expression is vague and difficult to understand. The author should carefully organize the language and reduce the use of similar sentences.
3、 The coordinates of all the figures in the manuscript have no units, and the legends in Figure 1 do not correspond to the curves.
4、 The probability density from 8:00 AM to 9:00 AM in Table 2 is decreased, which is inconsistent with the statement of Table 2 in the manuscript. And the forms of tables in the manuscript are not uniform.
The manuscript should be carefully revised and checked for grammatical errors to improve the quality of the manuscript.
Author Response
(1) There are many grammatical errors in the manuscript. The author should check and revise carefully.
- Grammar errors in the manuscript were carefully checked and corrected.
(2) The manuscript uses so many long sentences that the expression is vague and difficult to understand. The author should carefully organize the language and reduce the use of similar sentences.
- As you pointed out, I have carefully organized my language and reduced the use of long sentences and similar sentences.
(3) The coordinates of all the figures in the manuscript have no units, and the legends in Figure 1 do not correspond to the curves.
- As you pointed out, I added units to the figure coordinates, and also modified the legend of figure 1 to match the curves.
(4) The probability density from 8:00 AM to 9:00 AM in Table 2 is decreased, which is inconsistent with the statement of Table 2 in the manuscript. And the forms of tables in the manuscript are not uniform.
- From 8:00 am to 9:00 am, the probability density drops when vehicle operation and SMP1 rates are taken into account, but the probability density does not drop when SMP2 rates are considered, so it is judged to be more efficient. In addition, tables related to all probability density functions have been replaced with figures to improve readability.
Detailed modifications are marked in red in the attached file.
(5) The manuscript should be carefully revised and checked for grammatical errors to improve the quality of the manuscript.
- The quality of the manuscript was improved by carefully examining the manuscript as a whole and correcting grammatical errors.

Reviewer 3 Report
On page 2, the last line at the very bottom should be corrected. Similarly, on page 3, the last line at the very bottom should be corrected.
The results presented in this article provide a thorough understanding of how much the V2G strategy contributes to daily load flattening by charging and discharging electric vehicle batteries in workplaces and homes in Seoul. An interesting solution proposed in the article was a control algorithm containing a daily load curve for charging and discharging electric vehicles during the year, which is calculated by dividing it into uncontrolled mode and controlled mode.
Author Response
(1) On page 2, the last line at the very bottom should be corrected. Similarly, on page 3, the last line at the very bottom should be corrected.
- As you pointed out, I edited the last line on pages 2 and 3.
(2) The results presented in this article provide a thorough understanding of how much the V2G strategy contributes to daily load flattening by charging and discharging electric vehicle batteries in workplaces and homes in Seoul. An interesting solution proposed in the article was a control algorithm containing a daily load curve for charging and discharging electric vehicles during the year, which is calculated by dividing it into uncontrolled mode and controlled mode.
- As you explained, we established an electric vehicle charging and discharging strategy by dividing it into non-controlled mode and controlled mode for each scenario. And as a result, we calculated how much each mode contributed to daily load leveling for Seoul City and presented the effect.
Detailed modifications are marked in red in the attached file.

Reviewer 4 Report
This paper proposes the significant research direction of V2G strategy in the field of smart grids by studying the impact of electric vehicle charging and discharging behavior on the power grid. However, there are numerous issues in the paper, so we cannot accept your submission. The existing problems are as follows:
1. The abstract should be concise and concise, including four parts: research background and purpose, research methods, research results, and research conclusions. Currently, the logic of the abstract is chaotic and the content is lengthy.
2. The literature cited in the introduction is limited and has a long lifespan, which cannot represent the current research status in this field.
3. There is too little analysis of the content in Figure 1.
4. There is a formatting error in the content of Table 1.
5. Figures 2, 3, and 4 lack detailed textual descriptions.
6. The conclusion section should summarize research findings, answer research questions, and explain their significance and contributions. The conclusion of the current paper does not meet the requirements.
We found that your paper has a confusing structure, lacks clear hierarchy, and the key parts are not highlighted. This will bring great trouble to readers. Given the above issues, we are currently unable to accept your paper.
This paper demonstrates a solid level of proficiency in English language use. The author has demonstrated a strong grasp of grammar, syntax, and vocabulary, which enables clear and concise communication of their ideas. The paper's structure and organization are well-executed with appropriate use of headings, paragraphs, and transitions to guide the reader through the content.
However, there may be a few areas where the paper could be improved. The author may need to pay closer attention to word choice and avoid repetitive phrases or expressions. Additionally, while the paper demonstrates a strong ability to communicate ideas, there may be opportunities to strengthen the author's argument by drawing more explicitly on the research literature and showing how the author's work fits into the broader intellectual landscape.
Author Response
(1) The abstract should be concise and concise, including four parts: research background and purpose, research methods, research results, and research conclusions. Currently, the logic of the abstract is chaotic and the content is lengthy.
- As you pointed out, the abstract was edited concisely with the research background and purpose, methods, research results, and conclusions.
(2) The literature cited in the introduction is limited and has a long lifespan, which cannot represent the current research status in this field.
- The references cited in the introduction have been explained in more detail, and old papers have been excluded and new references have been added. However, please understand that there are not many references related to grid peak reduction using V2G, so many references cannot be reviewed.
(3) There is too little analysis of the content in Figure 1.
- As pointed out, we have presented an additional analysis for Figure 1.
(4) There is a formatting error in the content of Table 1.
- As pointed out, Tables 1-4 have been modified instead of graphs.
(5) Figures 2, 3, and 4 lack detailed textual descriptions.
- Added additional text descriptions for figures 2,3 and 4.
(6) The conclusion section should summarize research findings, answer research questions, and explain their significance and contributions. The conclusion of the current paper does not meet the requirements.
- In the conclusion section, we summarized the study findings and added and revised descriptions of important achievements and contributions.
(7) We found that your paper has a confusing structure, lacks clear hierarchy, and the key parts are not highlighted. This will bring great trouble to readers. Given the above issues, we are currently unable to accept your paper.
- As you pointed out, the structure of the thesis has been modified, and the content of key parts such as charge/discharge modeling and simulation analysis for each scenario have been reinforced and emphasized.
(8) This paper demonstrates a solid level of proficiency in English language use. The author has demonstrated a strong grasp of grammar, syntax, and vocabulary, which enables clear and concise communication of their ideas. The paper's structure and organization are well-executed with appropriate use of headings, paragraphs, and transitions to guide the reader through the content.
However, there may be a few areas where the paper could be improved. The author may need to pay closer attention to word choice and avoid repetitive phrases or expressions. Additionally, while the paper demonstrates a strong ability to communicate ideas, there may be opportunities to strengthen the author's argument by drawing more explicitly on the research literature and showing how the author's work fits into the broader intellectual landscape.
- As you pointed out, repetitive words have been avoided and additional explanations have been supplemented to better convey the idea of the thesis.
Detailed modifications are marked in red in the attached file.

Round 2
Reviewer 2 Report
The author has carefully revised the manuscript according to the review comments, and some details of the manuscript still need to be carefully checked. It is suggested that the manuscript be accepted after minor revision.
The English quality of the manuscript has been greatly improved.
Author Response
(1) The author has carefully revised the manuscript according to the review comments, and some details of the manuscript still need to be carefully checked. It is suggested that the manuscript be accepted after minor revision.
- In the introductory part of the thesis, the necessity and current status of V2G-related technologies were additionally explained. Specifically, in order to prepare for the grid situation caused by the expansion of electric vehicle supply, various studies and demonstrations on whether the power grid can meet the increasing power demand of electric vehicles using V2G technologies were explained. In addition, with the expected effect of V2G technology, it is a system technology that trades electricity in both directions between the electric vehicle and the power system. It not only transmits power from the power system to charge the electric vehicle, but also depends on the auxiliary service market or demand response signal in the power market during power peaks. The explanation that power generation and consumption efficiency can be increased because power can be sent back has been supplemented. Then, the related foreign art trends were analyzed. The specific technology trends were explained as follows. In other words, in the case of the United States, PJM-led mid-atlantic grid interactive car consortium and smart car consortium are conducting verification and evaluation of V2G technology, and in Korea, EV (Electric vehicle) in the smart transportation project of the Jeju smart grid demonstration complex explained that they are conducting V2G tests and demonstrations that send batteries back to the power system. However, in order for V2G technology to achieve its final goal, it is necessary to secure large-capacity, high-output, long-life and low-cost batteries, develop communication with the vehicle's main controller and customized services to determine battery charging/discharging time, linkage between AMI interface technology and V2G technology, and V2G technology. It was suggested that problems such as system permitting and stabilization measures for bi-directional power flow between operable vehicles and grid connection infrastructure should be addressed. Finally, in the conclusion section, it was stated that V2G strategy research according to changes in seasons and energy supply would be conducted in the future.
- English editing was performed through the MDPI English service.
Reviewer 4 Report
Dear author,
Thank you very much for submitting your article on the V2G strategy for flattening the daily load curve in Seoul, South Korea. We appreciate your research and insights on this important issue. However, after carefully reviewing your submission, we have had to decide to refuse to publish your article. Here are our rejection comments:
1. In terms of body structure, your article lacks a clear Logical framework. When presenting relevant information about the V2G strategy to readers, your article did not provide relevant background information or previous research results about the strategy. We suggest that you provide a background introduction at the beginning of the article to ensure that readers understand the basic principles and current applications of V2G technology.
2. In terms of data and statistics, your article did not provide sufficient supporting evidence. Regarding the viewpoint on how the V2G strategy affects the daily load curve in Seoul, South Korea, you simply expressed your belief without providing specific data analysis or model validation. We suggest that you cite relevant research or empirical data to enhance the credibility and reliability of your viewpoint.
3. Your article did not mention the challenges and limitations that the V2G strategy may face. It is crucial to understand and disclose the difficulties and limitations of emerging technologies in discussions. We encourage you to explore the feasibility of V2G technology and the technical, economic, and regulatory barriers considered for implementation in your article.
4. The conclusion of the article did not provide sufficient data and model validation. The author only concluded about the changes in the load curve by estimating the charging and discharging capacity of electric vehicles. However, in the absence of accurate data, these conclusions are questioned for their validity and reliability.
5. These conclusions did not fully consider other influencing factors. The change in load curve is not only influenced by the charging and discharging capacity of electric vehicles, but also by other factors such as weather, seasonal demand changes, and energy supply. The author did not fully discuss and analyze these factors, so these conclusions may only partially describe the actual situation.
Dear author,
My evaluation of the English proficiency of your paper is as follows:
The overall English expression in your paper is fluent, and the vocabulary and syntactic structure used are also accurate. I noticed that you used some specific terminology when describing the V2G strategy and related data, which shows your expertise in the field.
However, I also noticed some Syntax error and inaccurate expressions. When revising your paper, it is recommended that you carefully examine the following aspects:
Verb Tense: Ensure to use the correct tense to describe your research and results. Pay special attention to the transformation between the past, present and future tenses.
Grammar consistency: Maintain consistent grammar structure and sentence usage throughout the entire paper. Avoid using different grammatical forms in the same sentence or paragraph.
Article and article usage: ensure the correct use of Mass noun, plural nouns and related articles (a/an, the)
Punctuation: carefully check the use of Punctuation such as commas, periods, semicolons, etc. This helps to ensure the clarity and logic of sentences.
Despite these minor errors, overall, your English proficiency shows a high level. Please take the time to carefully review and revise these issues to ensure that the language quality of your paper is improved.
Author Response
(1) In terms of body structure, your article lacks a clear Logical framework. When presenting relevant information about the V2G strategy to readers, your article did not provide relevant background information or previous research results about the strategy. We suggest that you provide a background introduction at the beginning of the article to ensure that readers understand the basic principles and current applications of V2G technology.
- As you pointed out, a background explanation of V2G technology was introduced in the introduction. In other words, V2G technology is a system technology that trades power in both directions between an electric vehicle and the power system. It not only transmits power from the power system to charge the electric vehicle, but also transmits power back according to the auxiliary service market or demand response signal in the power market during power peaks. It can be expected to increase the efficiency of power production and consumption. In addition, when analyzing overseas technology trends, in the case of the United States, PJM-led mid-atlantic grid interactive car consortium and smart car consortium are proving and evaluating V2G (Vehicle to grid) technology, and in Korea, Jeju smart grid In the smart transportation project of the demonstration complex, V2G tests and demonstrations are being carried out to send EV (Electric vehicle) batteries back to the power system.
(2) In terms of data and statistics, your article did not provide sufficient supporting evidence. Regarding the viewpoint on how the V2G strategy affects the daily load curve in Seoul, South Korea, you simply expressed your belief without providing specific data analysis or model validation. We suggest that you cite relevant research or empirical data to enhance the credibility and reliability of your viewpoint.
-. As pointed out, in the case of vehicle movement data, which is essential in terms of related data and statistics, the average traffic volume for the year 2020 of the Seoul Metropolitan Police Agency was presented, and in the case of the existing load, data analysis was conducted based on the annual average sub-curve in Seoul, based on this presented the V2G strategy for 2030 and 2040. In addition, since traffic and power consumption patterns are different for each city, it is believed that using the actual traffic and power consumption patterns of Seoul in 2020 will increase the reliability of the thesis rather than citing empirical data.
(3) Your article did not mention the challenges and limitations that the V2G strategy may face. It is crucial to understand and disclose the difficulties and limitations of emerging technologies in discussions. We encourage you to explore the feasibility of V2G technology and the technical, economic, and regulatory barriers considered for implementation in your article.
-. As you pointed out, there are challenges and limitations that a V2G strategy can face. In the case of V2G technology, it is currently receiving a lot of attention along with efforts to spread electric vehicles. In this regard, there are considerations for future commercialization in various aspects. To summarize, it is as follows.
○ Securing high-capacity, high-output, long-life and low-cost batteries
○ Battery impact analysis and stable operation plan according to V2G operation
○ Standardization and performance improvement for V2G operating vehicles (communication network and protocol, battery capacity and voltage, high-efficiency bi-directional battery charger, etc.)
○ Development of communication with vehicle main controller and customized service to determine battery charge/discharge time (V2G operation technology without reducing user convenience)
○ Linkage plan between smart grid, AMI interface technology and V2G technology
○ System acceptance and stabilization plan for bidirectional power flow in charging infrastructure for electric vehicles such as rapid chargers and grid connection infrastructure in other V2G operable vehicles.
In addition, in the case of the United States, the share of ancillary service is 5-10% of the total electricity cost, which is a market worth 12 billion dollars per year, of which 80% of the cost is added by regulation and spinning reserve. there is. If V2G can replace the role of the grid operator, it can be easily guessed that it will occupy a huge new potential market.
(4) The conclusion of the article did not provide sufficient data and model validation. The author only concluded about the changes in the load curve by estimating the charging and discharging capacity of electric vehicles. However, in the absence of accurate data, these conclusions are questioned for their validity and reliability.
- In order to estimate the charging and discharging capacity of an electric vehicle, the mileage according to the battery capacity was prepared based on the British BERR report [Reference 19], and based on this, the V2G strategy was implemented to level the partial down curve in Seoul by the electric vehicle charging and discharging rate system. and analyzed its evaluation.
(5) These conclusions did not fully consider other influencing factors. The change in load curve is not only influenced by the charging and discharging capacity of electric vehicles, but also by other factors such as weather, seasonal demand changes, and energy supply. The author did not fully discuss and analyze these factors, so these conclusions may only partially describe the actual situation.
- As you pointed out, charging and discharging of electric vehicles is influenced by seasonal changes in demand and other external factors such as energy supply. However, it is realistically difficult to consider all factors in order to establish a V2G strategy that affects the leveling of the subsurface curve in the entire city of Seoul. Therefore, yearly demand change and electricity supply data were used to exclude seasonal variables, and research on changes according to seasons and energy supply will be conducted in the future.
(6). The overall English expression in your paper is fluent, and the vocabulary and syntactic structure used are also accurate. I noticed that you used some specific terminology when describing the V2G strategy and related data, which shows your expertise in the field. However, I also noticed some Syntax error and inaccurate expressions
- English editing was performed through the MDPI English service.
Round 3
Reviewer 4 Report
This paper aims to study the impact of charging and discharging of electric vehicles on the load of the power grid, and proposes a scheme to balance the load of the power grid through the V2G strategy. The research content of this article has certain significance, but the logic of the article is chaotic, the innovation of research methods is insufficient, and it does not meet the publication standards of the journal. The specific issues are as follows:
1.The urgency and necessity of the research content are not clearly described in the abstract section.
2. The introduction contains references with low quality, long lifespan, and limited quantity, which cannot provide a detailed explanation of the domestic and international research status of the research content.
3. There are too few textual explanations in Figures 2 and 3.
4.The description in the first paragraph of the conclusion is too lengthy and lacks emphasis.
At present, the level of this article has not reached the level of an academic paper. It is recommended that the author carefully revise it.
The English proficiency of this paper is relatively professional and technical, using complex terminology and sentence structures.However, it may be important to avoid excessive abbreviations and technical terms in order to better enable non professional readers to understand and master the content of the article. In addition, the length and structure of some sentences can be more concise and clear, making the article easier to read and understand.